# Selenium Compounds Affect Differently the Cytoplasmic Thiol/Disulfide State in Dermic Fibroblasts and Improve Cell Migration by Interacting with the Extracellular Matrix

**DOI:** 10.3390/antiox13020159

**Published:** 2024-01-26

**Authors:** Christine Kreindl, Sandra A. Soto-Alarcón, Miltha Hidalgo, Ana L. Riveros, Carolina Añazco, Rodrigo Pulgar, Omar Porras

**Affiliations:** 1Laboratory for Research in Functional Nutrition, Instituto de Nutrición y Tecnología de los Alimentos, Universidad de Chile, Santiago 7830490, Chile; ntakreindl@gmail.com (C.K.); miltha.hidalgo@inta.uchile.cl (M.H.); 2Department of Nutrition and Dietetics, Faculty of Health Sciences, Universidad Autónoma de Chile, Santiago 7500912, Chile; sandra.soto.alarcon@gmail.com; 3Laboratorio de Nanobiotecnología y Nanotoxicología, Departamento de Química Farmacológica y Toxicológica, Facultad de Ciencias Químicas y Farmacéuticas, Universidad de Chile, Sergio Livingston 1007, Santiago 8380492, Chile; riveros.ana@gmail.com; 4Laboratorio de Bioquímica Nutricional, Escuela de Nutrición y Dietética, Facultad de Ciencias para el Cuidado de la Salud, Universidad San Sebastián, General Lagos #1190, Valdivia 5110773, Chile; carolina.anazco@uss.cl; 5Laboratory of Genomics and Genetics of Biological Interactions, Instituto de Nutrición y Tecnología de los Alimentos (INTA), Universidad de Chile, Santiago 7830490, Chile; rpulgar@inta.uchile.cl

**Keywords:** HyPer biosensor, sodium selenite, selenium cysteine, selenium methionine, extracellular matrix, human dermic fibroblast, cellular migration

## Abstract

Deficient wound healing is frequently observed in patients diagnosed with diabetes, a clinical complication that compromises mobility and leads to limb amputation, decreasing patient autonomy and family lifestyle. Fibroblasts are crucial for secreting the extracellular matrix (ECM) to pave the wound site for endothelial and keratinocyte regeneration. The biosynthetic pathways involved in collagen production and crosslinking are intimately related to fibroblast redox homeostasis. In this study, two sets of human dermic fibroblasts were cultured in normal (5 mM) and high (25 mM)-glucose conditions in the presence of 1 µM selenium, as sodium selenite (inorganic) and the two selenium amino acids (organic), Se-cysteine and Se-methionine, for ten days. We investigated the ultrastructural changes in the secreted ECM induced by these conditions using scanning electron microscopy (SEM). In addition, we evaluated the redox impact of these three compounds by measuring the basal state and real-time responses of the thiol-based HyPer biosensor expressed in the cytoplasm of these fibroblasts. Our results indicate that selenium compound supplementation pushed the redox equilibrium towards a more oxidative tone in both sets of fibroblasts, and this effect was independent of the type of selenium. The kinetic analysis of biosensor responses allowed us to identify Se-cysteine as the only compound that simultaneously improved the sensitivity to oxidative stimuli and augmented the disulfide bond reduction rate in high-glucose-cultured fibroblasts. The redox response profiles showed no clear association with the ultrastructural changes observed in matrix fibers secreted by selenium-treated fibroblasts. However, we found that selenium supplementation improved the ECM secreted by high-glucose-cultured fibroblasts according to endothelial migration assessed with a wound healing assay. Direct application of sodium selenite and Se-cysteine on purified collagen fibers subjected to glycation also improved cellular migration, suggesting that these selenium compounds avoid the undesired effect of glycation.

## 1. Introduction

The prevalence of type II diabetes has increased by 50% in the last ten years [1]. One of the hallmarks of this disease is sustained hyperglycemia that leads to unspecific and extensive modification of lysine residues on proteins by Schiff base reactions, known as glycation [2]. Clinical complications, such as neuropathies, retinopathies, nephropathies, or diabetic foot ulcers, are related to the glycation degree in the circulating haemoglobin [3]. Cumulative glycation on extracellular matrix (ECM) proteins [4,5] exerts a negative influence on the endothelial function at the microvascular level [6], as one of the pathogenic mechanisms for peripheral organ failure observed in diabetic patients.

The collagen maturation at the endoplasmic reticulum of fibroblasts, one of the most abundant proteins in the ECM [7], requires ascorbic acid for enzymatic hydroxylation on lysine and proline residues [8]. Once in the extracellular space, collagen fibers become enzymatically cross linked by lysyl oxidase (LOX) to generate fibrils. Sustained circulating high glucose levels increase intracellular ROS production, likely by activating NADP(H) oxidase [9,10], and upregulate LOX [11,12], which together modify the mechanical properties of fibers [13,14]. Increased ECM stiffness, due to excessive crosslinking of fibers, decreases endothelial cell adhesion due to deficient anchoring of integrins with glycated proteins [15].

The antioxidant function of selenium proteins is well-supported by enzymes that regulate intra- and extracellular redox [16]. One of these selenoenzymes is glutathione peroxidase (GPx), whose primary function is to convert hydrogen peroxide into water by oxidizing the selenol (-SeH) group to selenic acid (-SeOH), which is reduced back by glutathione (GSH) [17]. A 50% decrease in the activity of GPx3 has been associated with lower levels of nitric oxide in subjects with early arterial thrombosis [18,19]. On the other hand, the thioredoxin reductase (TrxR) system is relevant for counteracting the oxidative stress generated by high glucose. For instance, fibroblasts from Txnrd1^−/−^ mice were more sensitive to pharmacological GSH depletion than wild-type at 25 mM glucose, indicating that TrxR1 is indispensable for cell survival and the elimination of H_2_O_2_ under high-glucose conditions [20]. Selenium can be incorporated into proteins as selenocysteine (SeCys) or selenomethionine (SeMet); for SeCys, which participates in the active site of GPx1, its incorporation is determined by the location of the UGA codon in the transcript [21] and several modifications on the anticodon loop in the transfer RNA [22]. Knocking out the tRNA gene for selenocysteine provokes general skin disorders, including aberrant hair follicle morphogenesis, hyperplastic epidermis, and progressive alopecia, in mice [23]. By adding ~30 nM sodium selenite to mesenchymal stem cells, Heo demonstrated increased proliferation of these cells and enhanced anti-inflammatory and angiogenic properties of exosomes in vitro and in vivo, which improved dermal fibroblast migration and wound healing [24]. Recently, other groups have demonstrated that topical application with inorganic selenium improved wound healing and collagen deposition in animal models [25,26,27,28,29,30]. However, it is still unclear if the selenium carrier, inorganic versus organic, is relevant to ECM production and its redox impact on living human fibroblasts. Selenium compounds, such as Na_2_SeO_3_, SeCys, and SeMet, have not been employed in cell culture for durations exceeding 72 h [31,32,33].

In this study, we evaluated whether long-term supplementation (ten days) with selenium amino acids (organic) or sodium selenite (inorganic) had an impact on the cytoplasmic oxidation/reduction balance of human dermic fibroblasts maintained at two glucose levels, 5 and 25 mM. By monitoring the HyPer signal, a fluorescent redox biosensor whose emission spectra change according to the disulfide/thiol state of a pair of cysteine residues, we corroborated that 25 mM glucose promoted a more oxidative cytoplasmic environment and that these fibroblasts produce a morphologically different ECM that offers reduced migration to microvascular endothelial cells. Surprisingly, all the treatments with selenium increased the steady-state oxidative tone and modified the redox responses of fibroblasts differently upon an oxidative challenge. Only selenium amino acids affected the ultrastructure of ECM fibers. Further evaluation of these ECMs by performing the wound assay with endothelial cells indicated that selenite and SeCys improved the quality of ECM, an effect related in part to interference with the glycation process.

## 2. Materials and Methods

### 2.1. Cell Culture

The CCD-1068Sk fibroblast cell line (ATCC^®^ CRL-2086™, Manassas, VA, USA) corresponds to normal mammary dermal tissue collected from a mastectomy procedure for mammary carcinoma. Cells were maintained in Minimum Essential Media (MEM) (Life Technologies, Carlsbad, NY, USA) supplemented with a 10% fetal bovine serum (Corning Inc., New York, NY, USA) and a 1% pen–strep solution (penicillin 10,000 U/mL–streptomycin 10 mg/mL, Biological Industries, Cromwell, CT, USA) under a humidified atmosphere at 37 °C and 5% CO_2_. The cultures were used between passages 10 and 17. Cells were maintained in normal (5 mM) (NG) or high (25 mM)-glucose (HG) concentrations for at least three passages. Ascorbic acid (155 µM) (Sigma-Aldrich, St. Louis, MO, USA) was added to the medium of full confluence cultures every day for ten days to promote collagen synthesis, which overall corresponds to 14 days in vitro [34]. The human dermal microvasculature cell line TIME (ATCC^®^ CRL-4025^TM^, USA) was maintained at 37 °C in M131 medium, enriched with Microvascular Endothelial Cell Growth Kit-VEGF (PCS-110-041, ATCC, Manassas, VA, USA) penicillin–streptomycin, and amphotericin under a humidified atmosphere with 5% CO_2_. Culture media was renewed every 2 to 3 days, and cells were used between passages 3 and 10. Viability assays were performed on confluent 12-well culture dishes to choose non-lethal concentrations of selenium compounds. A range of concentrations (0.25–50 µM) of sodium selenite (SS, Sigma-Aldrich, St. Louis, MO, USA), as well as selenomethionine and selenocysteine (SeMet and SeCys, respectively, Cayman Chemicals Company, Ann Harbor, MI, USA) was incubated for ten days. Viable cells were identified by observation with a Neubauer chamber using trypan blue (Sigma-Aldrich, St. Louis, MO, USA) [35].

### 2.2. HyPer Biosensor Imaging

HyPer is a fluorescent biosensor oxidized only by H_2_O_2_ that contains a pair of cysteines that, upon oxidation, induces a conformational change in its structure, resulting in altered spectral properties of the fluorescent domains [36]. This phenomenon is reversible and can be reduced by thioredoxin and glutathione reductase [37,38,39]. CCD1068Sk cells were seeded on glass coverslips and infected with Adeno-HyPer3 (at a 1:50 dilution). After 48 h, biosensor recordings were carried out using a Nikon TI Eclipse inverted epifluorescence microscope with a 40× objective [NA 1.3] (Melville, NY, USA). This microscope was connected to a xenon lamp and a monochromator (Cairn Research Ltd., Faversham, UK), allowing dual excitation at 420 and 490 nm. Emitted fluorescence was collected over 520 nm using a LongPass filter (Chroma Technology Corp., Bellows Falls, VT, USA). The biosensor signal was expressed as a 490/420 ratio. On the experiment day, the culture medium was replaced with KRH buffer (in mM: 140 NaCl, 4.7 KCl, 20 HEPES, 1.25 MgSO_4_, 1.25 CaCl_2_, pH 7.4) supplemented with 5 mM glucose, and the coverslip was mounted in an open recording chamber. The HyPer ratio signal was recorded every 20 s for a minimum of 15 min until a stable baseline was observed. Maximal oxidation of HyPer was induced by a 500 μM H_2_O_2_ pulse (5 min), which, after being washed out, allowed the kinetic evaluation of the biosensor signal recovery.

### 2.3. Analysis of Gene Expression

The fibroblast mRNAs were isolated with RNA-Solv^®^ Reagent (Omega Bio-Tek Inc., Norcross, GA, USA). Complementary DNA was generated with M-MLV reverse transcriptase (Promega, Madison, WI, USA) using oligo (dT) (Promega) using 1 µg of total RNA. Semiquantitative PCR reactions were performed using the following primers:

Glutathione peroxidase 1, *GPX1*, forward 5′-TATCGAGAATGTGGCGTCCC-3′ and reverse 5′-TCTTGGCGTTCTCCTGATGC-3′, PCR product size 143 bp; Thioredoxin reductase 1, *TXNRD1*, forward 5′-ATCATCATTGGAGGTGGCTCAG -3′ and reverse 5′-ACACATGTTCCTCCGAGACC-3′, PCR product size 134 bp; Aquaporin 1, *AQP1*, forward 5′-AGGCTACAAAGCAGAGATCGAC-3′and reverse 5′-CACCCTCTAAATGGCTTCATTC-3′, PCR product size 197 bp, Peroxiredoxin 1, *PRDX1*, forward 5′-GGGAACCTGGTTGAACCCC-3′ and reverse 5′-TGGCATAACAGCTGTGGCTT-3′, PCR product size 99 bp; Transforming Growth Factor β Receptor 1, *TGFβ-R1*, forward 5′-AAGTCATCACCTGGCCTTGG-3′ and reverse 5′-AGCAATGGTAAACCTGAGCCA-3′, PCR product size 242 bp; Transforming Growth Factor β Receptor 2, *TGFβ-R2*, forward 5′-ACGTTCAGAAGTCGGATGTGGAAA-3′ and reverse 5′-TCTCAGTGGATGGGCAGTCCTATT-3′, PCR product size 86 bp; Smooth muscle alpha-actin, *α-SMA*, forward 5′-GCCGACCGAATGCAGAAGGA-3′ and reverse 5′-TGCGGTGGACAATGGAAGGC-3′, PCR product size 190 bp; Thioredoxin 1, *TXN1*, forward 5′-CTTGGACGCTGCAGGTGATA-3′ and reverse 5′-TCCTGACAGTCATCCACATCT-3′, PCR product size 150 bp; 18S rRNA, forward 5′-CGCCGCTAGAGGTGAAATTCT-3′ and reverse 5′-CGAACCTCCGACTTTCGTTCT-3′, PCR product size 101 bp. The relative mRNA abundance was determined with real-time qPCR using an AriaMx Real-Time PCR System (Agilent Technologies, Santa Clara, CA, USA) with 400 ng of cDNA and a DNA Master SYBR^®^ Green I reaction mix in a final volume of 20 µL. All reactions were carried out in duplicate, including the negative control. The relative expression of each gene was relativized to the expression of the messenger of 18S rRNA and the equation described in [40].

### 2.4. Picrus Sirius Red Staining

CCD1068Sk cultures were decellularized with 0.5% *v*/*v* Triton X-100 in 20 mM NH_4_OH at 37 °C under gentle agitation for 15 min until cell bodies disappeared under visual inspection with bright field microscopy. After this, samples were fixed with 4% formaldehyde in PBS for 20 min at RT, then washed out twice and stained with 1 mg/mL Picrus Sirius Red (PSR) (Sigma-Aldrich, St. Louis, MO, USA) for another 20 min at RT. The excess of PSR was removed by washing the samples twice with 0.5% acetic acid (Merck, Darmstadt, Germany) and leaving the samples to dry. At least 18 images were taken for each condition and the experiments were repeated three times. Image analysis was performed with the Image J software (v.1.54f) [41].

### 2.5. Determination of Fiber Thickness with Scanning Electron Microscope (SEM)

After ten days of collagen synthesis, samples on 12 mm coverslips were decellularized and the cell-free surfaces were fixed for 12 h using 2% glutaraldehyde, 0.1 M sodium cacodylate, and 0.1 M sucrose in PBS (Sigma-Aldrich, St. Louis, MO, USA). Subsequently, the samples were washed with 0.1 M sodium cacodylate, followed by immersion in 1% osmium tetroxide (Sigma-Aldrich, St. Louis, MO, USA) in 0.1 M sodium cacodylate for 1 h. After this period, the solution was replaced with 1% tannic acid in 0.1 M sodium cacodylate solution for one hour more. Finally, the samples were dehydrated with ethanol and immersed in hexamethyldisilazane (Sigma-Aldrich, St. Louis, MO, USA) for 3 min [42]. After gold-sputter coating (10 nm) in an argon atmosphere (Sputter Coater Cressington TEDPELLA, 108, Redding, CA, USA), coated samples were observed with field emission scanning electron microscopy (FE-SEM, INSPECT-F50, Thermo Fisher Scientific, FEI, Waltham, MA, USA) using an accelerating voltage of 10 kV. Fiber thickness, number of branches, and crosslinking were analyzed using the open-access software Image J (v.1.54f) [41].

### 2.6. Cellular Migration Assay

An intra-chamber device (ab242285, Abcam, Cambridge, UK) was used for the wound-type assay with endothelial cells seeded on two types of surfaces: decellularized and purified collagen type I. To generate surfaces covered with collagen, 0.5 mg/mL of human collagen type I in PBS (Corning Inc., New York, NY, USA) was adjusted to pH of 7.0–7.2 and left for polymerization for 2 h at 37 °C. Glycation was achieved with 10 mM methylglyoxal (MGO) in the presence or absence of 1 µM of selenium compounds overnight at 37 °C in a humidified environment.

The endothelial cells were seeded on decellularized matrices or purified collagen with the intra-chamber device on the culture dish until 90% confluence was reached. At this time, the intra-chamber device was carefully lifted, and cells were gently washed twice with warm PBS. Images were taken under bright field microscopy at zero time and 8 h later. Then, the area of the existent gap was measured using Image J software (v.1.54f) [41].

### 2.7. Statistical Analysis

All data presented correspond to the average ± SE from at least three independent experiments. Comparisons of more than two experimental groups were analyzed with ANOVA with Bonferroni post hoc tests; for non-parametric tests, Kruskal–Wallis with Dunn’s post hoc tests; and averages between two groups, with Student’s *t*-test. Tests were performed on 3 independent samples and at least 3 technical replicates under the described experimental conditions. Data plotting and analysis were performed with Sigma Plot 12.0, Systat software, Inc. (San Jose, CA, USA)

## 3. Results

### 3.1. HyPer Responses in Human Dermal Fibroblasts Cultured in Normal and High Glucose

The HyPer biosensor was expressed in two groups of CCD-1068Sk cells, one maintained at 5 mM glucose (normal, NG) and the other at 25 mM glucose (high glucose, HG). Neither fibroblast set showed any evident morphology differences, but HG fibroblasts proliferated slower than NG fibroblasts. Figure 1A,B show the classic response of the biosensor when 500 µM H_2_O_2_ is added to the extracellular space; this stimulus provokes a rapid increase in the ratio signal until a plateau is reached. Then, after removing the oxidant agent, a recovery phase is noticeable, driving the signal back to basal values. NG fibroblasts elicited basal values significantly lower than those cultured in HG (Figure 1C). Despite this difference, the oxidation rate of HyPer induced by exogenous H_2_O_2_ was similar in both glucose conditions (Appendix A), which is consistent with similar levels of mRNA abundance for aquaporin-1 and peroxiredoxin-1, both involved in H_2_O_2_ permeability through the plasma membrane and intracellular transmission of H_2_O_2_-induced protein oxidation through the cytoplasm, respectively.

Another aspect we evaluated was the recovery rate that the HyPer biosensor undergoes spontaneously after oxidative pulse removal. This phenomenon is associated with the disulfide bond reduction activity present in the cytoplasm, which seems to be quite similar in NG fibroblasts, which recovered 83.7 ± 0.4%, compared with HG fibroblasts, which recovered 79.4 ± 0.7% (Figure 1D). To know the contribution of the thioredoxin system in reducing the oxidized biosensor, we added PX-12, an inhibitor of cytosolic thioredoxin-1, in the washout buffer for H_2_O_2_ removal. In the presence of this inhibitor, the biosensor HyPer only recovered 32 ± 1% in NG fibroblasts, whereas in HG fibroblasts, the recovery reached 73 ± 1% (Figure 1E), indicating that reduction of the disulfide bonds on the biosensor is mainly mediated by thioredoxin-1 in NG fibroblasts, a trait that is not present in HG fibroblasts. We found no differences in mRNA abundance for glutathione peroxidase-1, thioredoxin reductase-1, and thioredoxin-1 of NG and HG fibroblasts, suggesting that the difference in the efficiency of disulfide bond reduction in the cytoplasm of theses fibroblasts is determined in another level of redox enzyme regulation.

### 3.2. Redox Effects of Selenium Compounds on Dermic Fibroblasts

To select a non-cytotoxic concentration of selenium compatible with the entire period that fibroblasts need to generate ECM, we first evaluated cellular viability for three selenium compounds: sodium selenite, selenocysteine, and selenomethionine, for ten days after confluence. With this design, both fibroblast sets, NG and HG, started the selenium exposure with similar cellular densities. The analysis of dose–response curves rendered LC_50_ values for SS of 5.7 ± 0.7 µM in NG fibroblasts and 5.8 ± 0.6 µM in HG fibroblasts. In the case of selenium amino acids, the LC_50_ values obtained for SeCys in fibroblasts in NG was 16.2 ± 1 µM and in HG was 9.3 ± 1 µM. For SeMet, an LC_50_ of 7.9 ± 2 µM was obtained for fibroblasts in NG, whereas it was 7.9 ± 2 µM for fibroblasts maintained in HG (Appendix A). Accordingly, we decided to utilize 1 µM of each selenium compound in both glucose conditions based on the cell viability results and visual inspection of cellular morphology.

Contrary to what we expected, the sustained exposure with 1 µM of selenium, as either a salt or amino acids, augmented the steady-state ratio value of the biosensor in both conditions, normal and high-glucose (Figure 2A). Next, we observed that the HyPer biosensor responded faster to 500 µM hydrogen peroxide in NG fibroblasts treated with SS and SeCys than in non-treated fibroblasts; SeMet showed no significant effect on the biosensor oxidation rate (Figure 2B). In HG fibroblasts, only selenium amino acids significantly increased the rate of HyPer response upon exogenous H_2_O_2_ (Figure 2C) compared with non-treated fibroblasts. In this group, sodium selenite treatment showed no effect.

Furthermore, we analyzed the effect of the selenium compounds on the recovery rate of the biosensor. In NG fibroblasts, the treatments with selenium compounds did not affect the recovery rate of HyPer signal after H_2_O_2_-induced oxidation, except SeMet, which diminished this rate by half [non-treated: 17.2 ± 1 s^−1^; SeMet: 8.2 ± 1 s^−1^] (Figure 2D). In HG fibroblasts, we found that only treatment with SeCys accelerated the rate of the biosensor recovery from 13.9 ± 1 s^−1^ in non-treated to 19.2 ± 1 s^−1^ (Figure 2E). Intriguingly, when we tested the sensitivity of biosensor reduction to PX-12 in seleno-treated fibroblasts (Appendix A), we found in NG fibroblasts that the two selenium amino acids abolished the inhibitory effect that PX-12 exerts on the reduction of the oxidized biosensor, an effect that was absent with the sodium selenite (Figure 2F). A similar pattern was observed in HG fibroblasts cultured in the presence of selenium amino acids; however, sodium selenite sensitized these cells to PX-12, which significantly decreased the recovery degree of the oxidized biosensor (Figure 2F).

In summary, we could not observe a clear pattern of the redox impact of selenium on human fibroblasts. Nevertheless, the ratio signal of the HyPer biosensor at the baseline, which represents the thiol/disulfide balance at the steady state, was significantly increased by all types of selenium evaluated, independent of the glucose level in the culture conditions. Thus, we explored the possibility that the selenium treatment could modify the transcript levels of some cytoplasmic redox enzymes. Only treatment with selenomethionine increased the abundance of glutathione peroxidase-1 in both fibroblast sets. This compound also increased the abundance of thioredoxin reductase-1, thioredoxin-1, and aquaporin-1 in NG fibroblasts, but not in HG fibroblasts. In this latter group, SeMet was the only compound that significantly increased the abundance of peroxiredoxin-1 (Appendix A), which is in line with an accelerated oxidation of the biosensor. These results indicate that SeMet increased the abundance of two selenoenzymes only in NG fibroblasts, together with a remarkable increase in the mRNA levels for thioredoxin-1. However, these changes at the mRNA level were not accompanied by a more rapid disulfide bond reduction in the cytoplasm. In fact, the biosensor reported that NG fibroblasts treated with SeMet present a diminishment in the disulfide bond reducing activity in this cellular compartment.

### 3.3. Effects of High-Glucose Culture and Selenium Compounds on the Abundance, Thickness, and Crosslinking of Fibers Secreted by Dermal Fibroblasts

To observe the ultrastructure of the ECM secreted by fibroblasts, the cell bodies were removed, and the remaining material deposited on the surface was prepared for scanning electron microscopy and for collagen staining with Picrus Sirius Red. A comparative panel of representative images of extracellular collagen stained with PSR is shown in Figure 3A. After the analysis of the total surface covered by red stain, we observed a slight but significant decrease in the amount of collagen secreted by HG fibroblasts (from 22 ± 1% in NG fibroblasts to 15 ± 1% in HG fibroblasts, Student’s *t*-test). Regarding the effect of selenium compounds, treatment with SeCys induced a more abundant presence of collagen in NG fibroblasts (32 ± 4%), an effect that was not observed in HG fibroblasts (15 ± 1%). On the contrary, SeMet induced a dramatic diminishment in the collagen-covered extracellular surface of fibroblasts maintained in either normal or high glucose, 4.5 ± 0.4% and 4.9 ± 0.7%, respectively (Figure 3B).

Next, we examined the thickness, branching, and crosslinking of extracellular fibers with scanning electron microscopy (Figure 4A). By visual inspection of the SEM images, we could perform manual and precise measurements of 20 nm thickness for individual fibers, albeit our analysis was blind to thinner fibers. The population of fibers deposited by HG fibroblasts was thicker than those produced by NG fibroblasts [*p* < 0.0001, Student’s *t*-test] (Figure 4B). This morphological feature observed in high glucose was robust and not affected by selenium supplementation with SS and SeCys. Indeed, only treatment with SeMet, which dramatically decreases the collagen content in the ECM, also promoted the thickening of fibers in NG and HG fibroblasts.

The branching degree for each fiber was also evaluated. We only considered a maximum of 30 branches per fiber, an upper limit that included more than 90% of the observations; the data distribution can be found in Appendix A. With this approach, we noted that HG fibroblasts produced significantly more branched fibers than NG fibroblasts [*p* = 0.018, Student’s *t*-test] (Figure 4C). The higher branching induced by high glucose was not reversed by treatment with selenium compounds. On the contrary, branching was exacerbated by SeCys. Treatment with SeMet, on the other hand, promoted the branching of the whole population of fibers, including the group of fibers with more than 30 branches increasing from 5.4% in non-treated NG fibroblasts to 10% in those treated with SeMet. SeCys, on the other hand, increased this type of fiber population from 7.7% to 9.4% in HG fibroblasts. Likewise, SeMet increased the number of nodes in the fibers secreted by NG fibroblasts and SeCys produced a similar effect in HG fibroblasts (Figure 4D).

At this point, there is no clear pattern between the redox profiles and the ECM produced by fibroblasts exposed to the selenium compounds evaluated. Treatment with SeCys seems to confer a certain redox flexibility to fibroblasts by increasing the rate of protein oxidation upon an oxidative challenge as well as the rate of disulfide bond reduction. In this line, it is the only selenium compound that augmented the abundance of collagen in the ECM produced by NG fibroblasts. Treatment with SeMet, on the other hand, had a dramatic decrease in the collagen abundance in the ECM and is the only condition that diminished the rate of disulfide bond reduction.

### 3.4. Selenium Compound Treatment Improved Endothelial Cell Migration in Extracellular Matrix Synthesized by HG Fibroblasts and Purified Collagen Fibers Subjected to Glycation

To test if the changes observed in the extracellular matrix generated by fibroblasts exposed to selenium compounds had some relevance for tissue regeneration, we performed wound healing assays with TIME endothelial cells (ECs), which are derived from the microvascular bed of dermis (Figure 5A).

Our first finding indicates that ECs were less effective in closing the gap in the ECM secreted by HG fibroblasts compared to the ECM secreted by NG fibroblasts (65 ± 10% for NG; 39 ± 6% for HG). The treatments with SS and SeCys had no effect on the ECM produced by NG fibroblasts. Curiously, SeMet treatment negatively affected the quality of the ECM produced by NG fibroblasts, an effect that proved not be additive in the high-glucose condition. Apparently, the ECM produced by HG fibroblasts treated with SS and SeCys facilitates gap closure by endothelial cells (Figure 5B). To know if selenium compounds improve the ECM quality by direct interaction with collagen, we performed the wound assay on human collagen type I fibrils in the presence of the glycating agent methylglyoxal (Figure 5C). With this simplified in vitro approach, we found that ECs migrated less on collagen surfaces exposed to methylglyoxal than non-treated native collagen (native: 50 ± 2%; MGO: 36 ± 2%). Consistent with this finding, we also found similar effects for SeMet, which decreased endothelial migration on native or MGO-treated collagen. On the other hand, the polymerization of collagen in the presence of sodium selenite and SeCys avoids the MGO-induced migration impairment (Figure 5D). These pieces of evidence suggest that collagen glycation is important for endothelial migration and that selenite and SeCys somehow interfere with this chemical modification of collagen fibrils.

## 4. Discussion

The intracellular collagen biosynthesis and the extracellular maturation of collagen fibrils among other proteins are fundamental for ECM consolidation. The metabolism and redox status of fibroblasts, along with environmental factors such as glucose levels and ROS, might modulate the quality of this extracellular network. Here, we investigated if the intracellular redox state of fibroblasts, modulated by a high-glucose environment and selenium supplementation, was transmitted to the ECM in terms of collagen abundance, ultrastructural properties, and its quality for endothelial migration. In addition, the direct interaction between selenium compounds, the ECM, and purified collagen fibrils was evaluated.

### 4.1. Long-Term Exposure to Selenium Compounds and Its Antioxidant Effects

The redox status of cytoplasm in living fibroblasts was evaluated with real-time imaging of HyPer, a molecular tool that reports the thiol/disulfide bond balance and the kinetics of intracellular oxidation/reduction activity. High glucose in the culture condition promotes an oxidative intracellular environment by diverse mechanisms that include the inhibition of antioxidant defenses [43] and increased ROS production by mitochondria [44] or NOX [45]. Here, we observed an increase in the basal ratio signal of HyPer in HG fibroblasts compared to NG fibroblasts. With this approach, we also found that all the selenium compounds evaluated here (Na_2_SeO_3_, SeCys, and SeMet) augmented the oxidative tone in the cytoplasm independent of the glucose level in the culture. Kinetic analysis of the biosensor responses helped us to unveil remarkable differences between the redox effects induced by SeCys and SeMet. Although the selenium atom has the same valence in both molecules, only the SeCys treatment accelerated both the protein oxidation induced by an exogenous H_2_O_2_ pulse and the disulfide bond reduction rate in HG fibroblasts. SeMet, on the other hand, was the only compound that decreased the disulfide bond reduction rate in NG fibroblasts. Interestingly, only SeMet treatment increased the mRNA abundance for GPX1 and TXRD1, both selenoenzymes, along with a remarkable increase in the cytoplasmic thioredoxin-1, configuring an ideal antioxidant system for restoring protein oxidation. To conciliate this contradiction, we believe that 1 µM SeMet could be exerting a mild cellular stress, stronger than with high glucose or the other selenium compounds (Figure 2A), that triggers a transcriptional response of the mentioned redox enzymes but without an effect on the cellular antioxidant capacity. This pattern has been observed in F36P cells (a human myelodysplastic cell line), in which SeMet (500 µM, 48 h) induce an increase in intracellular ROS simultaneously with elevated levels of glutathione [46]. SeMet (10 µM) is also more efficiently incorporated than the closely related MeSeCys and is distributed in perinuclear regions of A549 cells after 24 h [47]. It also has been reported to be more effective to increase GPX activity than Se-methylselenocysteine hydrochloride and sodium selenite [48]. Recently, Noe et al. demonstrated in pancreatic cancer cells that another organic compound containing selenium (dibenzyl diselenide, 10 μM, 48 h) induced an increase in ROS in the cytoplasm and mitochondria, a phenomenon accompanied by a loss of mitochondrial integrity with the concomitant increment in the intracellular iron availability, leading to ferroptosis in cancer cells [49]. Mitochondrial ROS production and dissipation of electrochemical potential of this organelle have been reported in other mammalian cells exposed to selenite [50,51]. In the study of Wallenberg et al., selenite (5 µM) and GS-Se-SG (5 µM) increased the superoxide anion levels in HeLa cells, whereas Se-DL-cystine (100 µM) was better tolerated than the other compounds, and the authors recognized that these selenium compounds induced complicated biochemical patterns that bring these cells to death by different pathways [51]. Another study comparing the cytotoxic effect of selenium nanoparticles (SeNP) in five cancer cell lines found different sensitivities to SeNP affecting cell viability and proliferation, with an unclear pattern in the transcription of selenoenzymes and pro-apoptotic genes [52]. These observations, along with our findings, depict the complex nature of the tolerance/cytotoxicity balance that selenium presents. Many studies have reported a biphasic dose–response curve that shows that high doses of selenium become toxic in cellular [46,53,54] and animal [55,56,57] models. Ultimately, the cellular/animal outcome after selenium supplementation will depend on the physicochemical characteristics of the selenium compound used and the cellular phenotype under observation.

Another intriguing finding was the differential sensitivity to PX-12, an inhibitor of thioredoxin-1, that the fibroblasts showed in our experimental conditions. Binding studies of several chemical inhibitors for the viral cysteine protease Mpro from SARS-CoV-2, including PX-12, showed that this agent requires an interaction with cysteine residues at the catalytic site of this protease to exert the inhibition [58]. We believe that the high-glucose condition and the supplementation with selenium amino acids somehow shield or protect the cysteine in thioredoxin-1 from the PX-12. Since this protection was not observed with the inorganic selenite, we suggest that the difference in the valence state of selenium is relevant for biological interaction of PX-12.

### 4.2. Understanding the Impact of Glycation on Secreted Fibers: Exploring Selenium’s Role

ECM fibers exhibited different structural properties depending on the glucose conditions or selenium supplementation that fibroblasts face, although we found no clear pattern between the different redox profiles induced by selenium compounds and the characteristics of the EMC produced. Notably, treatment with SeCys increases the rate of thiol oxidation and disulfide bond reduction, conferring on the cytoplasm a more dynamic redox response upon oxidative challenge, for instance. The treatment with SeMet seems to impair the recovery of oxidized proteins in the cytoplasm, at least in NG fibroblasts. These partially opposite effects became evident when the collagen abundance in ECM was observed with PSR staining. SeCys increased the presence of collagen in ECM, whereas SeMet did the opposite. It is important to mention here that this technique does not allow the staining of other components of ECM, only collagen. Indeed, non-stained fibers are visible in the images provided in Appendix A, evidencing that ECM is composed of other fibers than collagen. By examining the secreted ECM using SEM, we could characterize all the fibers morphologically and corroborate that the high-glucose condition induced the secretion of thicker fibers by fibroblasts and that SeMet promoted the thickening of fibers even in HG fibroblasts, suggesting an additive effect.

Particularly, the thickening of collagen is a biological process that depends on the activity of LOX, an amino oxidase-type enzyme that promotes the crosslinking of tropocollagen fibers to form the collagen fibril by deamination of lysine residues necessary for the formation of covalent bonds between fibers [7]. There is evidence that high-glucose conditions increase the activity of NADPH oxidase with the concomitant production of H_2_O_2_ [59], events that can be linked to stimulation of LOX activity in human endothelial cells [11], which exhibit altered ECM characterized by higher crosslinking of collagen and elastin [10].

Consistent with the idea that SeMet induces oxidative stress on fibroblasts, thus affecting the quality of ECM by decreasing the collagen abundance and promoting fiber thickening, endothelial migration was also impinged. On the contrary, selenite and SeCys slightly improved the migration in ECM produced by HG fibroblasts. This pattern of effects was also observed in purified collagen fibers subjected to glycation, which indicated that sodium selenite and SeCys could interact with collagen at the molecular level. This type of interaction has been reported with selenium nanoparticles, which decreased the formation of advanced glycation end products by ribose on lactoglobulin protein [60]; this protection is probably due to selenium shielding amino groups, preventing the formation of the Schiff base [61].

Glycation on collagen fibrils has been associated with impaired cellular migration [62,63,64,65]. In most in vitro approaches, massive amounts of glucose or other glycan agents are used on purified collagen fibers. In this work, we studied the ECM generated by dermic fibroblasts cultured in 25 mM glucose. This concentration represents the hyperglycemia observed in diabetic patients [66] and is well-tolerated by cultured cells. Under this condition, the ECM covered less area than that secreted by NG fibroblasts, likely because the thicker and more cross-linked fibers produced by HG fibroblasts tend to be in groups composed of piled-up fibers. Whether this HG-promoted ECM arrangement has a different stiffness compared to the ECM generated by NG fibroblasts is unknown. However, atomic force microscopy measurements on plantar tissue obtained from diabetic patients indicate an increased stiffness compared with healthy patients [67]. High extracellular glucose leads to the establishment of an ECM with morphological and mechanical properties that impair cell migration. Selenium compounds improved the ECM generated by HG fibroblasts for endothelial cell migration; this positive effect was also observed in methylglyoxal-treated purified collagen fibers, at least with sodium selenite and selenocysteine. Unfortunately, the mechanisms behind this effect are puzzling, since selenomethionine, which shares the same valence on the Se atom as selenocysteine, does not affect migration.

### 4.3. Factors Inducing Fibroblastic Phenotypic Changes

In this study, we assessed the effects of selenium compounds on the abundance of deposited collagen and the ultrastructural characteristics of the extracellular matrix (ECM) in fibroblasts maintained under high glucose as an approximation of the hyperglycemic conditions observed in diabetics. In this context, ascorbic acid is crucial for promoting collagen and ECM synthesis [8]. A phenotypic shift from fibroblasts (CCD-1093Sk) to myofibroblasts, characterized by the expression of α-SMA and abundant secretion of collagen (tropocollagen), has been reported under 177 µM ascorbic acid and TGFβ supplementation on days 2 and 6. In this cellular model, the ascorbic acid in the media was a necessary condition for myofibroblast conversion [68]. The myofibroblast metabolism seems to be glycolytic instead of oxidative, with elevated mitochondrial ROS production, promoting fibrotic ECM synthesis with resistance to cellular apoptosis [69]. In our study, fibroblasts were exposed to high glucose and ascorbic acid for ten days after confluence. Although these conditions generated a more oxidative tone in the cytoplasm according to redox measurements with HyPer, we did not observe significant increases in the abundance of transcripts for α-SMA, TGFβ-R1, and TGFβ-R2, which are myofibroblast markers, either in fibroblasts with high glucose or in fibroblasts treated with selenium compounds (Appendix A).

High glucose levels per se have been associated with phenotypic changes in fibroblasts and implicated in delayed healing in diabetic ulcers. Ma et al. determined a higher abundance of senescence-associated proteins p21, p16, and p53, along with a lower abundance of dermal collagen in diabetic mice and patients. Dermal fibroblasts from diabetic subjects had diminished levels of pyruvate dehydrogenase kinase 4 (PDK4), an enzyme that regulates the glycolysis rate by phosphorylating the pyruvate dehydrogenase and the nuclear internalization of yes-associated protein 1 as well, which commands the expression of senescence markers. The authors demonstrated that restoring the levels of PDK4 with lentiviral infection in dermal fibroblasts not only avoided cellular senescence but also accelerated migration and wound closure [70]. Although we did not evaluate senescence markers in our HG fibroblasts, we observed several traits indicative of this condition, such as a decreased collagen production, a seven-fold decrease in proliferation along with three-fold lower protein content with respect to NG fibroblasts, and an increased oxidation of cytoplasmic proteins. Cellular H_2_O_2_ production increases with the number of cell passages, an observation reported in the primary culture of human skin fibroblasts by Ghneinm and Al-Sheikh [48]. The increase in cellular oxidative tone was accompanied by lower protein and DNA synthesis as the culture passaging increased. All these changes were accompanied by higher activity of three enzymes involved in glucose metabolism, phosphofructokinase, lactate dehydrogenase, and glycogen phosphorylase [48]. Our experiments were conducted between passages 10 and 17, which is a very narrow window of passaging, which helps to minimize biological variability and leaves the high-glucose condition as the main inducer of senescence in our dermal fibroblasts.

### 4.4. Selenium Compounds: Potential Topical Applications for Diabetic Wound Healing

While the redox effects of selenium compounds on NG and HG fibroblasts are unclear, we can assert that the ECM synthesized with selenite and selenocysteine is conducive to endothelial cell migration and prevents negative effects on migration caused by ECM glycation and purified collagen. From the perspective of migrating endothelial cells, a glycated collagen surface produces changes in the profile expression of integrins, increasing αVβ3 and decreasing α2 β1 and thus, promoting cellular adhesion [71]. These changes may be linked to impaired angiogenesis [72], which contributes to perpetuating dermic lesions in decompensated diabetic patients. Given the potential benefits of selenium compounds as promoters of migration under glycation and their ability to impede the formation of AGEs [65,66], they emerge as an alternative for treating chronic wounds such as diabetic foot ulcers. A transdermal patch with selenium and chitosan nanoparticles significantly reduced inflammation in 7 days in an animal model for wound healing study; this procedure also improved neovascularization and collagen formation by day 21 [25]. Similar results were observed by applying a hydrogel that consisted of selenium nanoparticles embedded in cellulose–gelatin, which increased collagen deposition, angiogenesis, and fibroblast activation in rats [29]. Combination with other compounds has also shown that selenium in a hydrogel structure and zinc can improve full-thickness histopathological indices of wound healing in pediatric lesions [73].

The current knowledge about how selenium supplementation modifies cell function is complicated. We expect to contribute to the field by investigating if the redox changes induced in dermic fibroblasts by selenium exposure are transmitted to the ultrastructure of fibers in the ECM and if these morphological changes are relevant for endothelial migration. Our findings indicate that although selenium, in all the forms evaluated here, promoted an increase in the oxidative tone in the cytoplasm of fibroblasts, this stereotyped redox shift was not transferred with a clear pattern to the morphology of fibrils. However, by testing migration with endothelial cells, we found that both selenite and SeCys improved cellular migration in ECM produced by HG fibroblasts and glycated collagen fibrils. Consistently, SeMet, the only compound that impaired migration on both the mentioned surfaces, was also the only one that diminished the collagen abundance in ECM produced by dermic fibroblasts independent of glucose levels in culture conditions.

Overall, our data are useful for identifying SeCys over SeMet as the best choice to promote collagen production in dermic fibroblasts and improved endothelial migration in glycated ECM.

## 5. Conclusions

The effects of high glucose on cellular redox status are not mitigated by selenium compound treatment, nor are the structural changes in fibers secreted by fibroblasts. Nevertheless, the selenium compounds SeCys and SS transform the extracellular matrices secreted by fibroblasts cultured in high glucose into a more transitable surface for microvascular endothelial cells.

## Figures and Tables

**Figure 1 antioxidants-13-00159-f001:**
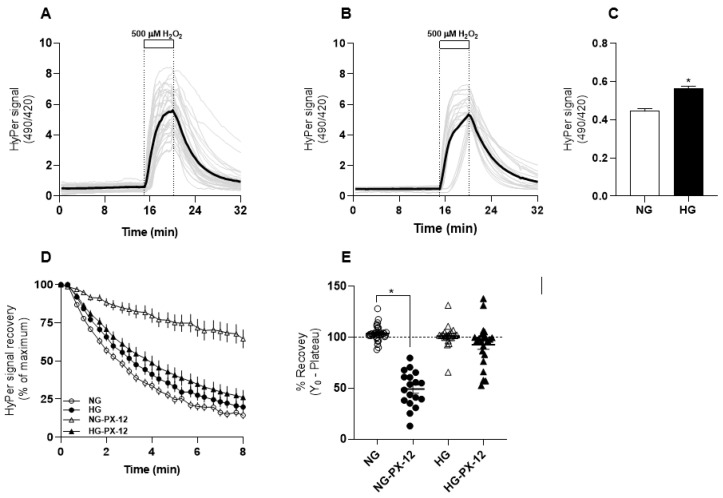
Comparison of redox responses of HyPer-expressing human fibroblasts cultured at 5 and 25 mM glucose. In (**A**), the time courses of 37 single-cell recordings of HyPer obtained from cells maintained at 5 mM glucose (NG) are shown with grey lines and the average of this group is depicted with a black line. A white box and dotted line indicate the moment and period of exposure to 500 µM H_2_O_2_. (**B**) is similar to A, but here, 23 single-cell recordings come from to fibroblasts maintained at 25 mM glucose (HG). In both cases, imaging data were collected from four independent experiments. (**C**) compares baseline values of the biosensor obtained from NG fibroblasts (empty bar) and HG fibroblasts (filled bar). Basal values were collected from 5–6 random fields from each coverslip, a procedure repeated in four independent experiments. This procedure rendered 129 and 133 cells overall for NG and HG, respectively. The data correspond to the average ± SE, and the asterisk represents the significant difference between the groups according to Student’s *t*-test. In (**D**), HyPer signal recovery was obtained right after H_2_O_2_ removal, when the biosensor is oxidized and presents the maximal signal. Data correspond to the average ± SE of 37 cells for NG (empty circles) and 23 for HG (filled circles). These data sets present the average ± SE of 24 recordings for NG-PX-12 (empty triangles) and 23 cells for HG-PX-12 (filled triangles). (**E**) compares the recovery observed in the four groups from (**D**) with data corresponding to average ± SE. The asterisk indicates significant differences between the groups according to one-way ANOVA with Bonferroni post hoc tests.

**Figure 2 antioxidants-13-00159-f002:**
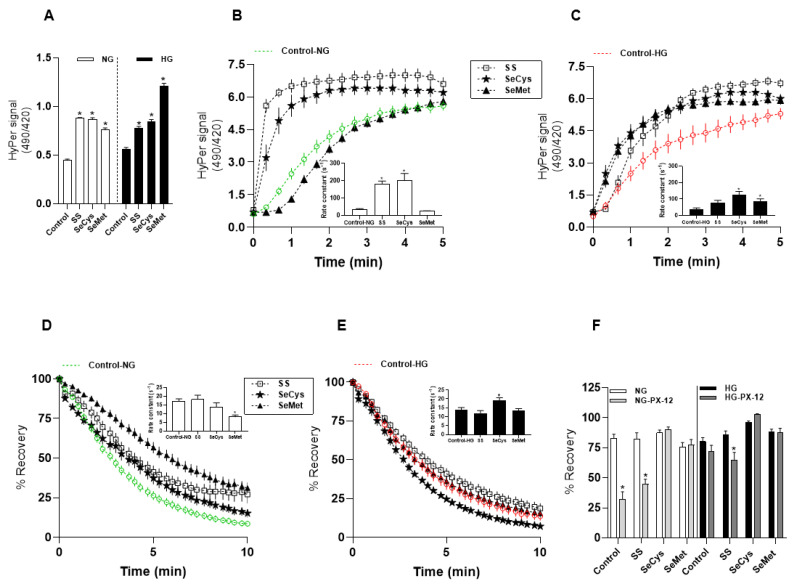
Redox alterations induced by the treatment with selenium compounds in human fibroblasts. (**A**) Comparison of baseline values obtained from HyPer-expressing fibroblasts maintained at 5 (empty bars) and 25 mM glucose (filled bars) and treated for ten days with sodium selenite (SS, 1 µM), selenocysteine (SeCys, 1 µM), or selenomethionine (SeMet, 1 µM). For this, baseline values from 136 to 170 cells were collected from four independent experiments. The data correspond to the average ± SE, and the asterisks represent significant differences between the treated groups versus non-treated according to one-way ANOVA with post hoc Bonferroni tests. (**B**) the time course of HyPer signal increase in NG fibroblasts upon hydrogen peroxide exposure. The data correspond to the average ± SE of 37 cells for NG (green), 29 cells for NG-SS (empty squares), 23 cells for NG-SeCys (filled stars), and 37 cells for NG-SeMet (filled triangles) conditions. The inset graph shows the rate constants (*b*) obtained from data fitting to the function $HyPer\;ratio=\left(Maximal-baseline\right)*e^{bt}$. The asterisks indicate significant differences between the treated and non-treated groups according to Kruskal–Wallis tests with Dunn post hoc tests. (**C**) Identical to B, but here time courses of HyPer signal increase were obtained from 23 HG fibroblasts (red), 23 cells for HG-SS (1 µM) (empty squares), 28 cells for HG-SeCys (1 µM) (filled stars), and 33 cells for HG-SeMet (1 µM) (filled triangles) conditions. (**D**) the time courses of HyPer signal recovery obtained from NG fibroblasts (green) and subjected to treatments with SS (empty square), SeCys (filled stars), and SeMet (filled triangles). The data are expressed as the percentage of the maximal signal obtained from the oxidized biosensor and correspond to the average ± SE. The inset graph shows the rate constants (*b*) obtained from fitting data from each single-cell recording to an exponential decay function. (**E**) similar to (**D**), with the difference that here the data correspond to HG fibroblasts (red), with the same symbols for the treatments. The asterisk indicates a significant difference within the NG and HG groups according to Kruskal–Wallis tests with Dunn post hoc tests. (**F**) the comparison of the PX-12 effects on the recovery (1 µM, gray bars) with those that are untreated. In the left panel are the NG fibroblasts treated or not with selenium compounds. In the right panel, we have the percentage of recovery obtained from HG fibroblasts. The data correspond to average ± SE and the asterisks indicate significant differences within the groups according to one-way ANOVA with Bonferroni post hoc tests.

**Figure 3 antioxidants-13-00159-f003:**
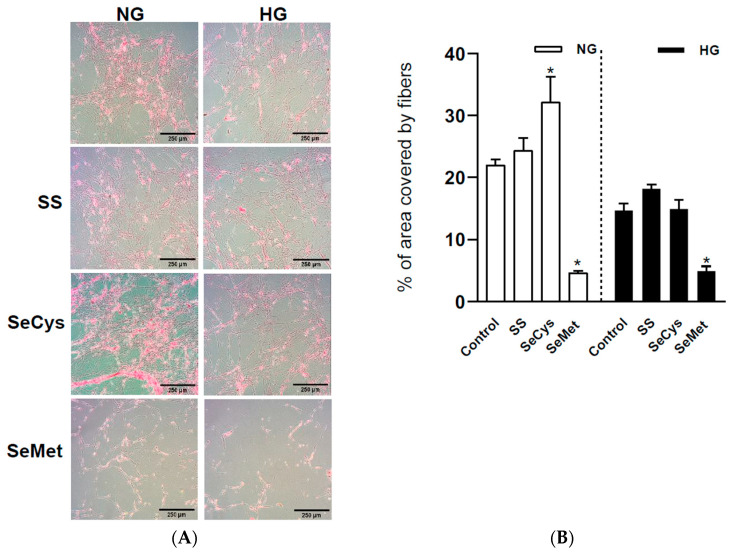
Effect of selenium compound treatments on the abundance of extracellular collagen secreted by CCD1068Sk cells. In (**A**), the panel of images shows the staining with Picrus Sirius Red on the surface that CCD1068Sk cells left after decellularization. The black bar in the lower right corner of each image represents 250 µm. NG and HG fibroblasts were treated with 1 µM sodium selenite (SS), selenocysteine (SeCys), or selenomethionine (SeMet). In (**B**), the area covered by the staining with Picrus Sirius Red was quantified by following a sequential step that included background subtraction/RGB stack/threshold in the ImageJ software. Experiments were conducted in triplicate and repeated three times. The data are expressed as average ± SE. The asterisks indicate significant differences within the NG (empty bars) and HG (filled bars) groups with different selenium compounds according to Kruskal–Wallis tests with post hoc Dunn tests.

**Figure 4 antioxidants-13-00159-f004:**
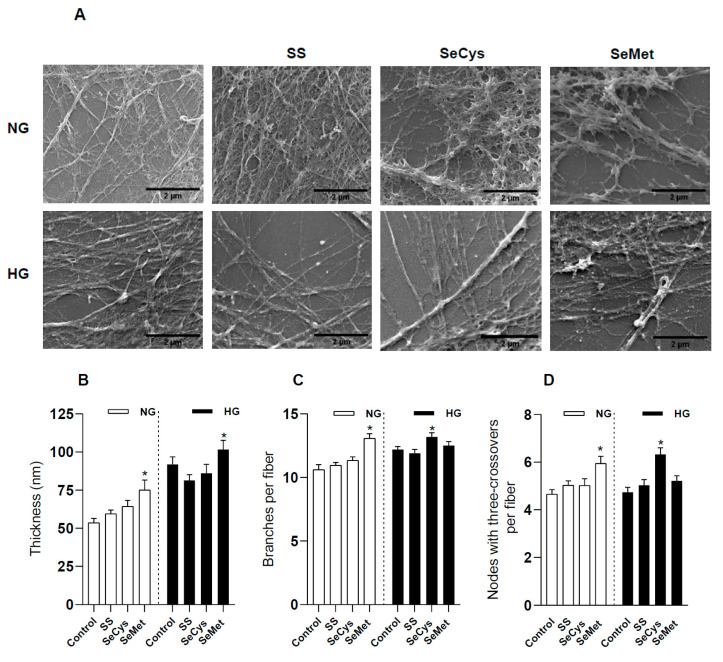
Effect of high glucose and selenium compounds on thickness, numbers of branches, and crosslinking of fibers secreted by dermal fibroblasts. CCD1068Sk cells were maintained under the collagen synthesis protocol for 10 days in normal glucose (NG, 5 mM) and high glucose (HG, 25 mM) in the presence of 1 µM selenium compounds: sodium selenite (SS), selenocysteine (SeCys), and selenomethionine (SeMet). Subsequently, the samples were decellularized and fixed. (**A**) a set of representative images taken with a scanning electron microscope at 50,000×. The black bar in the lower right corner of each image represents 2 µm. (**B**) refers to a comparison of thickness, (**C**) shows the number of branches, and (**D**) indicates crosslinking nodes of at least 50 fibers per condition [NG, empty bars; HG: filled bars]. The number of branches was obtained after skeletonizing the images and only branches associated with an intersection were considered. At least 9000 intersections were used for this analysis. The data represent the average ± SE from two to three independent trials. The asterisks indicate significant differences within NG or HG groups according to Kruskal–Wallis tests with post hoc Dunn tests.

**Figure 5 antioxidants-13-00159-f005:**
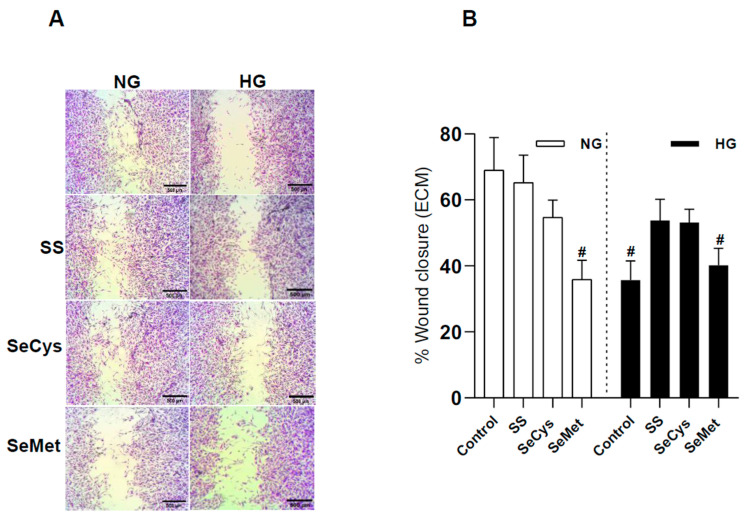
Sodium selenite and SeCys but not SeMet restore the quality of ECM produced by HG fibroblasts and glycated collagen fibrils for endothelial migration. (**A**) shows a panel of representative images of wound assays performed on TIME cells taken 8 h after growing in ECM produced by NG and HG fibroblasts in the presence of 1 µM sodium selenite (SS), selenocysteine (SeCys), or selenomethionine (SeMet). For scale, the black bar at the corner of the image represents 500 µm. In (**B**), the quantification of the gap area filled by TIME cells (p6-10). The wound closure percentage is the average ± SE of 18 observations from three independent experiments. The hash indicates significant differences in the treatments compared to the NG group according to Kruskal–Wallis tests with post hoc Dunn tests. In (**C**), similar to (**A**), TIME cells were seeded on purified collagen type I treated or not with methylglyoxal (MGO) and selenium compounds. (**D**) shows the quantification of the gap area filled by TIME cells (p6-10). The data represent the average ± SE from 18 observations from three independent experiments. The hash indicates significant differences in the treatments compared to the native group according to Kruskal–Wallis tests with post hoc Dunn tests.

## Data Availability

Data is contained within the article.

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
