# Peer review of "Selenium Compounds Affect Differently the Cytoplasmic Thiol/Disulfide State in Dermic Fibroblasts and Improve Cell Migration by Interacting with the Extracellular Matrix"

_antioxidants, 2024, doi:10.3390/antiox13020159_

Round 1

Reviewer 1 Report

Comments and Suggestions for Authors

The authors report that selenium compounds affect differently the cytoplasmic thiol/disulfide state in dermic fibroblasts and improve cell migration by interacting with the extracellular matrix. This study focuses on whether these selenium compounds effectively ameliorated the perturbations induced by culturing fibroblasts at high glucose levels regarding cytoplasmic redox, extracellular matrix and glycation on collagen fibers. These results indicate that cytoplasm protein oxidation was sensitive to selenium compound supplementation. Selenium amino acids increased the sensitivity to protein oxidation, and only Se-cysteine increased the disulfide bond reduction in fibroblasts maintained in high glucose. The article is interesting. However, some points of the manuscript also should be improved.

1.   It is better for authors to highlight the result of these experiments. It is so hard to find the valid information in the whole article.

2.   The lines in the figures in 3.2 look like they were drawn by authors themselves, which is not rigorous enough. What`s more, it even affects the reader's judgment about the data. Please use relevant annotations to mark them in the original layer.

3.   Please check the annotations for each figure.

4.   The authors need to add the control group in the figure 2.

5.   It is better to add more data and expressions about the NG group and native group in 3.5.

6.   Please shorten the abstract to highlight the results of this work.

7.   Please carefully check the manuscript for writing and gramma.

Comments on the Quality of English Language

  Please carefully check the manuscript for writing and gramma.

Author Response

Dear reviewer,

We are thankful for the comments and have improved the whole ms. Your indications facilitated the improvement process: exhaustive revision of each figure and legend, including the revision of the abstract and results, and final English revision for the whole ms (see the attached document). Below, you will find the actions that we took for each point you mentioned (in blue letters):

1. It is better for authors to highlight the result of these experiments. It is so hard to find the valid information in the whole article.

R:  We agree with this perception. In the result section, we emphasised the findings at the end of each group of results. For instance, redox evaluation of fibroblasts with HyPer and ECM and native fibers evaluation in result sections, have a parragraph to summarize the findings.  

2. The lines in the figures in 3.2 look like they were drawn by authors themselves, which is not rigorous enough. What`s more, it even affects the reader's judgment about the data. Please use relevant annotations to mark them in the original layer.

R: Thanks for the observation. The lines on the graphs 2B, 2C,2D and 2 E, correspond to the behavior of HyPer biosensor in control conditions for NG (green) and HG (red). We originally decided to put only a line as a reference for the condition without selenium, avoiding the overload in the plots of figure 2. In this new version, control data for NG and HG is now with symbols, green (NG) or red (HG) lines.

3. Please check the annotations for each figure.

R: We check carefully all the annotations in this new version. 

4. The authors need to add the control group in the figure 2.

R: Right. Control condition was indicated in the insets as NG or HG. Now, we denominated control-NG and control-HG, which were used for statistical comparison.

5. It is better to add more data and expressions about the NG group and native group in 3.5.

R: Following the recommendation, results obtained in NG-fibroblast were added in a composed plot (Figure 5B) and, the corresponding text was added to the ms (lines 416-418). For the native condition, we fix the issue in a similar manner. The figure was modified (Figure 5D) and text lines were added in the ms (lines 425-427). 

6. Please shorten the abstract to highlight the results of this work.

R: Thanks. An improved version of the MS is presented. 

7.   Please carefully check the manuscript for writing and gramma.

R: The writing and grammar in this new version have been reviewed by a certified English Teacher.

Reviewer 2 Report

Comments and Suggestions for Authors

In the present work, the authors study the impact of selenium compounds on the cytoplasmic oxidation/reduction balance of human dermic fibroblasts in the presence and absence of high glucose medium concentration. The topic of this research article is quite novel; however, several points need extensive revision before reconsidering it. In the introduction, the literature gap that the present study aims to cover should be emphasized. The manuscript lacks clear research question. What was the aim of the experiments? I completely fail to see the reasoning behind this study. Another major problem of this work is the absurdly high concentrations of H2O2. 500 microM is very high concentrations and most cells are probably dead at this experimental point. I usually use a 50-fold lower concentration to induce and analyze the effect of oxidative stress in living cells in a culture model, as do other authors. Returning to hydrogen peroxide - it is not clear in what context it is used. Once the authors write about cytoplasm protein oxidation (abstract) and then about  cytoplasmic oxidation/reduction balance. The authors must provide results of cellular viability for H2O2. Moreover, Induction of oxidative stress using hydrogen peroxide is a very poor reflection of the phenomena occurring in the body of diabetics and involving oxidative stress. The main role there is seen in mitochondrial dysfunction. It is unclear how CL50 was calculated, whether empirically or by fitting experimental data with logistic (or other type) equations. Typically, EC50 values are defined through data approximation with logistic equations, yielding CL50 (at inflection points of activity against varied concentrations of the compounds). As  authors stated they use GraphPad Prism 8 software for data plotting and analysis. Thus all parameters of CL50 should be presented. Authors stated that they used SYBR Green in qRT-PCR analysis. The problem with this dye is that it binds equally to the specific product of the amplification reaction and to any non-specific products and primer dimers as well. Provide gel electrophoresis pictures containing clear products of all qRT-PCR experiments. Provide proof that beta actin expression (used as control for RT-qPCR) was not affected by experimental design.

Reviewer 3 Report

Comments and Suggestions for Authors

In the submitted work, the authors present the effect of three selenium compounds (sodium selenite, selenocysteine, selenomethionine) in the antioxidant activity in cells and in the improvement of extracellular matrix-cell interaction. The authors attempt to study the effect of the selenium compounds in cell culture for durations of 72 h. Interestingly, the authors have found that the selenium compounds increase the migration of endothelial cells.

It should be noted that the biological effects of selenium have been previously investigated in detail in various in vivo models as the author report in their introduction (e.g. topical application of selenium compounds increases would healing, deleting of tRNA for selenocysteine). These previous studies significantly reduce the novelty of the current study.

There are some points that needed to be addressed by the authors. In many cases the authors fail to provide a description following their experiments that causes difficulties in reading the manuscript.

Specifically,

Line 77 silencing… the Reference 23 is knocking out Sec tRNA not silencing

Line 78 tRNA transfer aminoacids do not encoding for

Figure 1 shows 600 uM hydroxen peroxide but in the text it is written 500 uM

Line 298-299 why 1 mM selenium induced higher base line?

Line 312-313 what is the significance of this?

Line 318-319 what is the cause of this difference?

Based on Figure 4, SeMet shows a highly interconnected network of fibrils. However, this is not shown in Figure 3 stained for collagen. How do the authors interpret this effect? What is the nature of the fibrils in Figure 4?

Line 419 sodium selenite and selenocysteine dissipated this effect, from the data on Figure 5D it appears that only sodium selenite exerts an effect.

Overall, what is the connection of their findings with diabetes that is mentioned in the introduction? Do the authors suggest that selenium supplementation would be beneficial for chronic wounds in diabetic patients? And how? Topical administration? As mentioned the beneficial effect of selenium in wounds is already reported.

Author Response

Dear Reviewer 3:

Thanks for all your comments and suggestions. We are glad to bring an improved version of the ms that we expect to show a better description of the results. We added new areas of discussion regarding fibroblast senescence and transformation in myofibroblast. We worked from the abstract to the discussion and, finally, passed for a process of English correction by an external service. You can find this new version in the attachment, together with figures and legends, to facilitate the revision.

Bellow, we addressed each specific point you share with us, our responses are in blue letters:

Line 77 silencing… the Reference 23 is knocking out Sec tRNA not silencing

Line 78 tRNA transfer aminoacids do not encoding for

R: thanks for the correction. The corrected text is between lines 77-79 of the new MS version

Figure 1 shows 600 uM hydroxen peroxide but in the text it is written 500 uM

R: 500 µM H2O2 was the concentration used. We believed that pdf resolution or pdf conversion on the platform did something with the little numbers on the plot. In the new version, we put numbers a little bit larger.

Line 298-299 why 1 mM selenium induced higher base line?

R: Could near-lethal concentrations, as we used here, trigger some pro-oxidative mechanisms? Recently, Noe et al. demonstrated in pancreatic cancer cells that an organic compound containing selenium (dibenzyl diselenide) induced an increment in ROS at the cytoplasm and mitochondria, a phenomenon accompanied by a loss of mitochondrial integrity, increases in iron availability, that together contributes to sensitize cancer cells to cell death. In addition to the mechanism of selenium–induced toxicity described above, another study showed that selenium nanoparticles induced differential calcium responses in four human cancer cells. Moreover, these nanoparticles promoted the expression of pro-apoptotic and redox enzyme genes with different profiles, indicating that the cellular impact of selenium depends on cell phenotype.

We added a full paragraph in the discussion section abording the increased ROS production observed in cellular models exposed to different types of selenium. (Lines 482-507).

Line 312-313 what is the significance of this?

As suggested in some reports, selenium supplementation sensitizes cells to cell death by apoptosis but not by necrosis, offering a potential therapeutic for cancer treatment. In this sense, the cellular uptake of SeMet is more adventitious than Se-methyl selenocysteine and selenite, an observation achieved in A549 (human lung cancer) and Caco-2 cells. Although our focus was not on cancer therapeutics, we expanded our discussion to the cytotoxic mechanisms of selenium in several cellular models. Please see the lines 479-507.

Line 318-319 what is the cause of this difference?

First, we observed that both selenium amino acids avoid the PX12 effect on the spontaneous reduction of disulfide bonds of cytoplasmic HyPer expressed in NG-fibroblasts probably by interfering with the critical cysteine residues located at the catalytic site of Thioredoxin-dependent enzymes. Binding studies of several chemical inhibitors, including PX12, on the viral cysteine protease Mpro from SARS-CoV-2 showed that the action mechanism for inhibition requires that PX12 binds cysteine residues on the catalytic site of this protease. We believe that selenium amino acids can shield or protect the cysteine from the PX12 effect; since this protection was not observed with the inorganic selenite, we suggest that the difference in the valence state of selenium is relevant for biological interaction. This explanatory argument for our observation is now between lines 508-516 in the discussion section.

Based on Figure 4, SeMet shows a highly interconnected network of fibrils. However, this is not shown in Figure 3 stained for collagen. How do the authors interpret this effect? What is the nature of the fibrils in Figure 4?

In both figures, the ECM deposits obtained from decellularized matrices were fixed with PFA and dehydrated. However, there are fundamental differences between the visualization with PSR collagen staining and the electronic microscopy. First, the magnification in both cases is orders of magnitude different; in PSR, the spatial resolution is poor and the best resolution photos were taken at 400X magnification, whereas in SEM, the magnification is 50.000X. Another point is related to the nature of the PSR staining, which only allows the visualization of acidic motifs on collagen fibers. In SEM, on the other hand, the electronic beam scans all the fibers present in the ECM (collagen, fibronectin, laminine, etc.). We added another set of photos with better spatial resolution in which the non-stained PSR fibers can be noted on the background, supporting our explanation (supplemental figure 5). This issue was also included in the discussion, lines 527-533.

Line 419 sodium selenite and selenocysteine dissipated this effect, from the data on Figure 5D it appears that only sodium selenite exerts an effect.

We understand the point. We mentioned that SS and SeCys dissipated the effect of MGO because the incubation with both compounds counteracted the MGO effect on migration, losing statistical significance. Now, the figure has been expanded to show the effect of selenium compounds on migration in the two groups: native and MGO-treated fibres. In this new data display, it can be observed that SeMet decrease migration on collagen fibers independently of MGO-induced glycation. To ask if SS and SeCys were effective in counteracting the loss of migration on MGO-treated fibrils, we ran an ANOVA test comparing the conditions within the MGO group that indicated that only SS was capable of restoring native fibrils from the MGO-induced glycation. See the new Figure 5 and the improved description between lines 418-432.

Overall, what is the connection of their findings with diabetes that is mentioned in the introduction? Do the authors suggest that selenium supplementation would be beneficial for chronic wounds in diabetic patients? And how? Topical administration? As mentioned the beneficial effect of selenium in wounds is already reported.

We expect to contribute to the field by investigating if the redox changes induced in dermic fibroblasts by selenium exposure are transmitted to ultrastructure of fibers in ECM and if these morphological changes are relevant for endothelial migration. Our findings indicated that although selenium, in all the forms evaluated here, promoted an increase in the oxidative tone at the cytoplasm of fibroblasts, this stereotyped redox shift was not transferred with a clear pattern in the fibrils morphology. However, by testing migration with endothelial cells, we found that both selenite and Secys improved cellular migration on ECM produced by HG-fibroblasts and glycated collagen fibrils. Consistently, SeMet, the only compound that impaired migration in both mentioned surfaces, was also the only one that diminished the collagen abundance in ECM produced by dermic fibroblasts independently of glucose levels in culture conditions.

Overall, our data identify SeCys over SeMet as the best choice to promote collagen production in dermic fibroblasts and improve endothelial migration on glycated ECM.

We took the point and added this topic at the end of our discussion, understanding the potential application on wounds and if our finding served to select one selenium compound over another. See discussion lines 633-647.

Finally, we appreciate your observations since they were a guide to substantially improve our ms. In the attachment, you will find the whole ms, including supplemental figures and their respective legends.

Reviewer 4 Report

Comments and Suggestions for Authors

1. While the study provides valuable data on the redox effects of selenium and its impact on ECM, more detailed mechanistic insights could strengthen the study. This includes exploring the molecular pathways affected by selenium treatment and how they specifically contribute to changes in ECM properties and cell migration.

2. Comparison with Other Antioxidants: Including a comparison with other well-known antioxidants might offer a broader context to understand the unique or superior effects of selenium compounds.

3. Beyond the biosensor and ECM studies, investigating how selenium affects the phenotype of fibroblasts (e.g., changes in fibroblast to myofibroblast transformation) under high glucose conditions could provide additional insights into the wound healing process in diabetic conditions.

4. Your study investigates three forms of selenium, but it might be beneficial to delve deeper into the specific roles and mechanisms of each form (sodium selenite, selenocysteine, and selenomethionine). This could involve exploring their individual impacts on fibroblast function and ECM properties in high glucose conditions to understand their distinct biological activities.

5. Investigating how selenium compounds influence key signaling pathways in fibroblasts, such as the TGF-β, MAPK, or PI3K/Akt pathways, could provide a more nuanced understanding of the molecular changes induced by selenium in high glucose environments.

6. Considering the role of cellular senescence in diabetic wound healing, examining how selenium compounds affect the senescence of fibroblasts in high glucose conditions could provide valuable insights. This could involve looking at markers of cellular senescence, such as p16, p21, or SA-β-gal activity.

7. While your study examines the ultrastructure of the ECM, further analysis of the composition of ECM proteins (like collagen types, fibronectin, laminins) in response to selenium treatment under high glucose conditions could provide deeper insights into the quality of ECM being produced and its implications for wound healing.

Comments on the Quality of English Language

Moderate editing of English language required

Author Response

Dear Reviewer 4,

I am grateful for all your observations on behalf of Christine and me. Particularly for calling our attention to cellular senescence and fibroblast transformation to myofibroblasts, processes very relevant for wound healing.  We are glad to send an improved version of our ms together with figures and legends (in the attachment). Below, we show the reply to each point you brought up in blue letters:

1. While the study provides valuable data on the redox effects of selenium and its impact on ECM, more detailed mechanistic insights could strengthen the study. This includes exploring the molecular pathways affected by selenium treatment and how they specifically contribute to changes in ECM properties and cell migration.

We agreed. Our study is descriptive and lacks any mechanism that helps to explain how selenium affects the production and functionality of ECM produced by fibroblasts. Although at the beginning of this study, we thought that selenium supplementation would be followed by a more reducing cytoplasmic environment with higher expression of redox enzymes, we obtained the opposite panorama; we observed fibroblasts with higher oxidative tone at their cytoplasm. This is the first time that the redox impact of selenium supplementation has been monitored with a fluorescent biosensor; most of the previous reports examine the impact of selenium on stressed cells or animal models to then determine the ROS levels through chemical-based fluorescent dyes. Despite the robust redox response of fibroblasts to selenium supplementation, we observed opposite effects in collagen production; whereas SeCys promoted the secretion of ECM, SeMet diminished the collagen abundance in ECM, even when both compounds are organic and correspond to selenium amino acids. How are the selenium-induced redox changes in fibroblast transmitted to ECM? The chosen concentration of 1 µM Selenium was maybe within the tolerance limit and triggered the typical increase in ROS production observed in several cancer cell lines, in which the loss of mitochondrial electrochemical potential and ROS generation have been reported as part of the mechanisms involved. These perturbations have been associated with the fibroblast-to-myofibroblast transition that promotes excessive ECM production in mice lungs.

The discussion was re-redacted in this new version of the ms, including the mechanisms involved in selenium cytotoxicity and differences in selenium uptake, among others. Most of them were placed in the 4.2 section of the discussion.  

2. Comparison with Other Antioxidants: Including a comparison with other well-known antioxidants might offer a broader context to understand the unique or superior effects of selenium compounds.

In this study, all three selenium compounds generated a more oxidative tone in the cytoplasm of fibroblasts, an effect observed under both glucose levels used in the culture conditions. This effect was unexpected because we worked with 1 μM of selenium, a sub-lethal dose of selenium.

We agree with the idea of doing a comprehensive screening with antioxidant molecules with different structures and action mechanisms (NAC, DTT, or phenolic compounds) to generate different redox profiles that help to identify which of them fits better with the production of an ECM that facilitates endothelial migration. However, this task is an interesting issue for another research we should do soon. Sincerely, thanks.       

3. Beyond the biosensor and ECM studies, investigating how selenium affects the phenotype of fibroblasts (e.g., changes in fibroblast to myofibroblast transformation) under high glucose conditions could provide additional insights into the wound healing process in diabetic conditions.

Thanks for this comment. For this new ms version, we evaluated mRNA for α-sma/TGFß receptors 1 and 2 by qPCR. Those new results are presented in the supplementary Figure 4. In addition to this, we also added a new batch of cDNA to increase the number of samples for qPCR of redox enzymes. According to the new results and our experience culturing this cell line of dermic fibroblasts, we enriched the description of the results at lines 225-229, 240-243 and 300-312. Consistently, we modified the discussion adding this topic between lines 462-507.

We also added a new topic in the discussion section (4.3, from line 472), touching on myofibroblast transformation and cellular senescence.  

4. Your study investigates three forms of selenium, but it might be beneficial to delve deeper into the specific roles and mechanisms of each form (sodium selenite, selenocysteine, and selenomethionine). This could involve exploring their individual impacts on fibroblast function and ECM properties in high glucose conditions to understand their distinct biological activities.

Good point. In this new MS version, we have touched on specific particularities of the selenium compounds regarding their cellular uptake (Lines 482-498) and toxicity found by other researchers. Moreover, we include evidence of selenium interference with glycation (lines 555-571).

5. Investigating how selenium compounds influence key signaling pathways in fibroblasts, such as the TGF-β, MAPK, or PI3K/Akt pathways, could provide a more nuanced understanding of the molecular changes induced by selenium in high glucose environments.

We agree with this. Unfortunately, we could only add new detections for myofibroblast markers this time. Further experiments should include MAPK and PI3K/Akt pathways since we observed different proliferation rates between NG and HG fibroblasts. We know now that selenium compounds exhibit cytotoxicity by altering mitochondria and, therefore, the metabolism. We believe that all this data is enough for another ms. Thanks for your generous comments, anyway.    

6. Considering the role of cellular senescence in diabetic wound healing, examining how selenium compounds affect the senescence of fibroblasts in high glucose conditions could provide valuable insights. This could involve looking at markers of cellular senescence, such as p16, p21, or SA-β-gal activity.

This is an interesting point. After an exhaustive revision of our data, we realized that differences in proliferation were evident between fibroblasts maintained in 5 or 25 mM glucose. Those changes include less protein production and increased oxidative tone at the cytoplasm. HG-fibroblasts produce less ECM than NG-fibroblasts. All these changes depict senescence traits, and the issue was placed between lines 590-611 at the new topic of discussion 4.3, "Factors Inducing Fibroblastic Phenotypic Changes".

7. While your study examines the ultrastructure of the ECM, further analysis of the composition of ECM proteins (like collagen types, fibronectin, laminins) in response to selenium treatment under high glucose conditions could provide deeper insights into the quality of ECM being produced and its implications for wound healing.

Absolutely. We agree with the necessity to work hard on the characterization of ECM. The term “quality” should be addressed from its protein composition, as you suggested, and also directed to know better about the oxidative modifications that collagen fibrils offer to endothelial integrins. 

Again, thanks for your valuable comments.

Round 2

Reviewer 4 Report

Comments and Suggestions for Authors

The author has solved my doubts and improved the manuscript completely.

I recommend that this manuscript be published in this journal.

Comments on the Quality of English Language

Minor editing of English language required.